# A new no-report paradigm reveals that face cells encode both consciously perceived and suppressed stimuli

Janis Karan Hesse[1,2]*, Doris Y Tsao[1,2]*

[1]Division of Biology and Biological Engineering, Computation and Neural Systems, Pasadena, United States; [2]Howard Hughes Medical Institute, Pasadena, United States

**Abstract** A powerful paradigm to identify neural correlates of consciousness is binocular rivalry, wherein a constant visual stimulus evokes a varying conscious percept. It has recently been suggested that activity modulations observed during rivalry may represent the act of report rather than the conscious percept itself. Here, we performed single-unit recordings from face patches in macaque inferotemporal (IT) cortex using a no-report paradigm in which the animal's conscious percept was inferred from eye movements. We found that large proportions of IT neurons represented the conscious percept even without active report. Furthermore, on single trials we could decode both the conscious percept and the suppressed stimulus. Together, these findings indicate that (1) IT cortex possesses a true neural correlate of consciousness and (2) this correlate consists of a population code wherein single cells multiplex representation of the conscious percept and veridical physical stimulus, rather than a subset of cells perfectly reflecting consciousness.

*For correspondence:
jhesse@caltech.edu (JKH);
dortsao@caltech.edu (DYT)

Competing interests: The authors declare that no competing interests exist.

## Introduction

Having conscious experience is arguably the most important reason why it matters to us whether we are alive or dead. The question which signals in the brain reflect this conscious experience and which reflect obligatory processing of input regardless of conscious experience is a central puzzle of neuroscience. For example, activations in the retina may correlate with the conscious percept of flashing light but are arguably entirely driven by physical input, much of which never evolves into a conscious percept. Another driver of neural activity that can be confounded with signals related to conscious perception is report. Recently, it has been suggested that brain regions may correlate with conscious perception simply because they are driven by the active report of it (*Aru et al., 2012*; *Block, 2019*; *Block, 2020*; *Boly et al., 2017*; *Frässle et al., 2014*; *Koch et al., 2016*; *Overgaard and Fazekas, 2016*; *Panagiotaropoulos et al., 2020*; *Safavi et al., 2014*; *Tsuchiya et al., 2016*; *Tsuchiya et al., 2015*).

A paradigm known as binocular rivalry is useful for distinguishing responses related to conscious perception from those driven by obligatory processing of physical input (*Blake et al., 2014*; *Tong et al., 2006*): when two incompatible stimuli such as a face and an object are shown to the left and right eyes, respectively, one does not perceive a constant superimposition of the two but rather an alternation between face and object, even though the physical input is fixed (*Figure 1a*). Since these alternations are internally generated, they cannot be attributed to pure feedforward processing of external input.

In previous studies, researchers trained monkeys to report their percept during binocular rivalry by releasing a lever. They found that the proportion of cells modulated by the reported percept increases along the visual hierarchy, with 20% of cells showing modulations in V1 (*Leopold and*

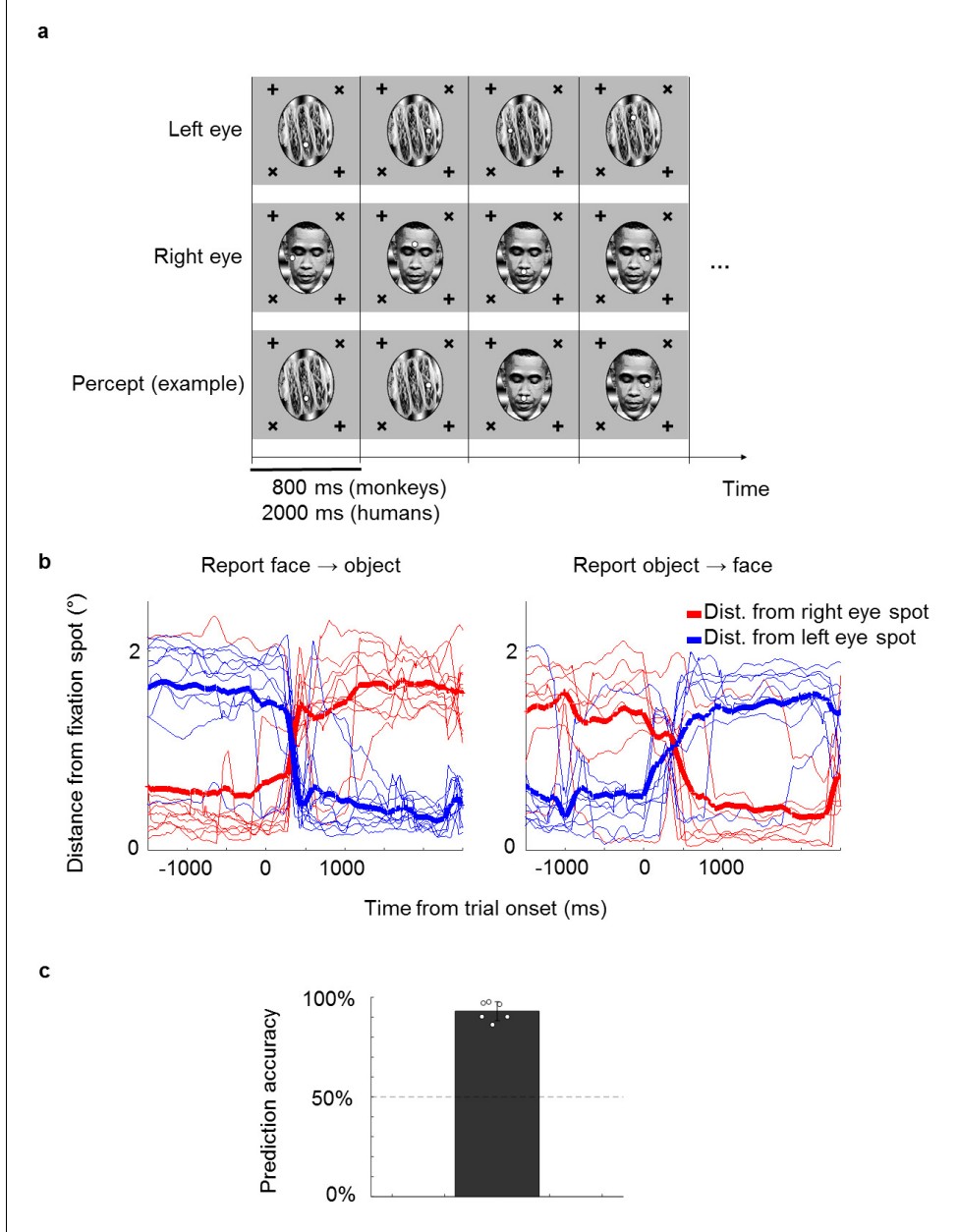

**Figure 1.** A novel no-report paradigm. (**a**) Illustration of binocular rivalry stimuli used in the paradigm. Four example trials are shown. Each trial was presented continuously for 800 ms without blank period between trials. The first and second rows show stimuli in the left and right eyes, respectively. If different stimuli are shown to the left and right eyes, as in this example, one's percept will spontaneously alternate between the two, as shown in the example perceptual trajectory in the third row. Stimuli in each eye contained a fixation spot at one of four possible positions. (**b**) Example eye traces from a human subject. Red and blue traces show the distance of the eye position from the fixation spot in the right and left eyes, respectively. Thick lines show the average. Traces are aligned to the onset of a trial where the subject reported that the percept switched from face to object (left) or object to face (right). (**c**) The bar plot shows the average proportion of trials where the percept inferred matched the percept reported by button press. White circles show accuracies of individual subjects. We inferred that a subject was perceiving face or object if the subject fixated on the face fixation spot (i.e., fixation spot in the eye of the face stimulus) or object fixation spot (i.e., fixation spot in the eye of the object stimulus), respectively, for at least half of the trial.

*Logothetis, 1996*) compared to 90% of cells showing modulations in inferotemporal (IT) cortex (*Sheinberg and Logothetis, 1997*). Using functional magnetic resonance imaging (fMRI), *Tong et al., 1998* found that the human fusiform face area responds to reported perceptual switches. Using single-unit recording, *Gelbard-Sagiv et al., 2018* found that the activity of neurons in the human medial temporal lobe and frontal cortex is also modulated by the reported percept.

Although binocular rivalry dissociates the conscious percept from physical input, an important confounding factor remains. In all studies cited above, the monkey or human subject always actively reported their percept by a motor response. Thus it is possible that the observed neuronal activations were due to the act of report itself, including introspection, decision making, and motor action accompanying report, rather than a switch in conscious percept. This concern was emphasized in an fMRI experiment by *Frässle et al., 2014* who compared modulations in the brain with and without active report. Many of the modulations observed in higher-level brain regions such as the frontal lobe disappeared when subjects did not actively report perceptual switches.

To infer the subject's percept in the absence of report, Frässle et al. used two no-report paradigms that depended on pupil size and optokinetic nystagmus, respectively. To exploit pupil size, they presented stimuli with different brightness in the two eyes, causing the subject's pupil size to vary according to the dominant percept's brightness. To exploit optokinetic nystagmus, they presented gratings moving in opposite directions in the two eyes, causing the subject's eye position to reflexively follow the direction of the dominant grating. Therefore, the conscious percept could be inferred by reading out pupil size and drift of eye position, respectively.

These no-report paradigms allow accurate prediction of the subject's percept but are not free of confounds themselves (*Overgaard and Fazekas, 2016*). First, pupil size is known to correlate with arousal, surprise, attention, and other confounding factors (*Bradley et al., 2008*; *Hoeks and Levelt, 1993*; *Preuschoff et al., 2011*). Second, when optokinetic nystagmus is applied to moving non-grating stimuli such as natural objects that drive IT cortex, there will be confounding physical stimulus differences. For example, the dominant stimulus that is smoothly pursued by the subject's eyes will tend to be stationary on the subject's fovea and optimally modulate IT areas with foveal biases, while the non-dominant stimulus will be more eccentric and have increased motion velocity. Moreover, optokinetic nystagmus is still present in monkeys in which the conscious percept is diminished due to anesthesia with low doses of ketamine (*Leopold et al., 2002*).

Here, we introduce a new no-report paradigm that relies on active tracking of a fixation spot, unlike the reflex-based paradigms mentioned above. In this fixation-based paradigm the subject is required to maintain fixation on a jumping spot, a task that many animals in vision research are already trained to perform. While following the fixation spot, subjects view either unambiguous, monocular stimuli physically switching between a face and an object, or a binocular rivalry stimulus that switches only perceptually. For the binocular rivalry stimulus, a fixation spot is shown to each eye at different positions on the screen. Thus, when perceiving a face in the left eye, the subject will generally perceive only the fixation spot in the left eye and saccade to it, ignoring the fixation spot in the right eye. In this way, the subject's percept can be inferred from eye movement patterns without active report.

In a second innovation, we performed electrophysiological recordings using a novel 128-electrode site Neuropixels-like probe that allowed us to measure responses from large numbers of cells simultaneously. This allowed us to address for the first time the extent to which neural activity is modulated by conscious perception in *single trials*. *Sheinberg and Logothetis, 1997* reported that 90% of IT cells are modulated by conscious perception. However, a fact that has been largely overlooked is that the response modulations found in that study during the rivalry condition were clearly smaller than those in the physical condition. It is possible that the decrease arose due to incorrect reporting of the percept by the monkey on some trials, and cells were modulated just as strongly by perceptual as by physical alternations. However, the decrease could also have been due to a more interesting possibility: mixed selectivity of cells for the conscious percept and the suppressed stimulus on single trials in the rivalry condition. In other words, it is possible that single cells encode both the conscious percept and the suppressed stimulus during rivalry. Inter-trial averaging confounds these two possibilities. To distinguish them, it is critical to compare perceptual vs. physical response modulations for single trials.

To explore correlates of conscious perception, we targeted recordings to macaque face patches ML and AM. The macaque face patch system constitutes an anatomically connected network of

regions in IT cortex dedicated to face processing (*Chang and Tsao, 2017*; *Grimaldi et al., 2016*; *Hesse and Tsao, 2020*; *Tsao et al., 2006*) and has served as an archetypal system for understanding object recognition in IT in general (*Bao et al., 2020*). To date, most response properties of cells in the face patch network can be explained using a feedforward framework without invoking conscious perception. For example, the functional hierarchy of this network, with increasing view invariance as one moves anterior from ML to AM (*Freiwald and Tsao, 2010*), can be explained by simple feedforward pooling mechanisms (*Leibo et al., 2017*). The representation of facial identity by cells in face patches through projection onto specific preferred axes can also be explained by feedforward mechanisms (*Chang and Tsao, 2017*).

Here, we explore activity in the face patch network using a binocular rivalry paradigm in which neural activity modulation is difficult to explain by feedforward filtering processes, since the stimulus remains unchanged. The hierarchical and feedback-rich organization of the face patch network (*Freiwald and Tsao, 2010*; *Grimaldi et al., 2016*) makes it a ripe testbed to examine the neural circuits underlying construction of conscious visual experience beyond feedforward filtering of visual input. It has been postulated that the fundamental architecture of the cortex is a predictive loop in which inference guided by internal priors plays a key role in determining what we see (*Rao and Ballard, 1999*). One explanation for binocular rivalry is that it arises as a consequence of such predictive coding, reflecting a high-level prior that two objects cannot occupy the same space (*Hohwy et al., 2008*).

We recorded from fMRI-identified face patches ML and AM in two monkeys using high channel-count electrodes, while we inferred the animals' conscious percept through the no-report paradigm described above. We found that large proportions of cells in both face patches (57% in ML and 73% in AM) encoded the conscious percept even without active report. Population activity of perceptually modulated cells was modulated more weakly during rivalry than during physical stimulus transitions in single trials. Nevertheless, we could reliably decode the dynamically changing conscious percept from activity in single trials. Surprisingly, we could also decode suppressed stimuli using activity from the same cells, indicating that single cells multiplex information about the conscious percept and the suppressed stimulus. These findings suggest that the neural correlate of consciousness within IT cortex resides in a population code rather than a subset of cells perfectly reflecting consciousness, and different linear readouts can decode either the consciously perceived or the suppressed stimulus from the same population.

## Results

We first confirmed that it is possible to correctly infer a subject's conscious percept using a fixation-based no-report paradigm through a behavioral experiment in humans. We presented binocular rivalry stimuli consisting of a face (e.g., Obama) in the right eye and a non-face object (e.g., a taco) in the left eye, causing the percept to stochastically alternate between the two (*Figure 1a*). Each of the stimuli contained a fixation spot that jumped to one of four possible locations every trial. Trials were 2000 ms long and contained no blank period, that is, stimuli were presented continuously. If subjects fixated at the fixation spot presented in the right eye on a given trial, we inferred that they perceived the face and vice versa for the object. To verify that the percept of face or object could be inferred from fixations, we instructed six naïve human subjects to perform the fixation task while simultaneously reporting their conscious percept with button presses. On trials where the percept switched, subjects also switched the fixation spot they were following (*Figure 1b*). We were able to infer which image the subjects were consciously perceiving with accuracies ranging from 86% to 98% across subjects (average: 93%, *Figure 1c*).

We next used the same method in monkeys to infer their conscious percept while recording from face patches ML and AM in IT. Importantly, the two monkeys in this study had never been trained to report their percept. They had previously been trained to maintain fixation on a spot (presented binocularly) and learned to perform the new task within 1 or 2 days, respectively (maintaining fixation on a spot for at least 80% of all trials). Since the monkeys were so adept at the task, we set the trial length to 800 ms (compared to 2000 ms in humans); this allowed higher temporal fidelity in determining the animal's percept. We presented two types of stimuli: in the 'physical' condition, unambiguous monocular stimuli were physically switched between face and object. In the 'perceptual' (binocular rivalry) condition, the same face and object were continuously presented to the right and left eyes, respectively, so any changes in percept were internally generated. To account for

individuals' eye dominance, we balanced the contrasts of the stimuli in the two eyes so that the monkey followed both fixation spots equally often in the rivalry condition. After balancing, median dominance durations were 7.2 s for faces and 7.2 s for objects across the two monkeys. Similarly, in human subjects, median dominance durations were 8 s for faces and 10 s for objects as estimated from fixation patterns, and 8.1 s for faces and 8.3 s for objects as estimated from reports. We inferred switches during rivalry when monkeys behaviorally switched from following the fixation spot in one eye to following the fixation spot in the other eye (example eye traces, *Figure 2a*, top). Spike rasters from an example ML cell showed a stronger response after switches from face to object compared to switches from object to face (*Figure 2a*, bottom; rasters aligned to onset of trials in which a switch occurred). *Figure 2b* compares average response time courses to physical vs. perceptual switches in two example cells, one from ML and one from AM. Both cells responded more strongly to a physically presented face than object. Importantly, in the binocular rivalry condition the response of both cells was also higher when the monkey perceived a face (as inferred by its eye movement) than when the monkey perceived an object. Since the physical stimulus was constant in this condition, the response reflected the monkey's conscious percept of a face and not just the physical input.

We recorded a total of 348 cells in ML and 210 cells in AM that were selective, i.e., they showed a significant difference between face and object in the physical switch condition ($p<0.05$, two-sided two-sample t-test). Since we recorded from face patches, most cells showed stronger responses to the physically presented face stimulus. Importantly, most cells kept their preference in the binocular rivalry condition (*Figure 3*). In face patch ML, 57% (200/348) of cells were significantly modulated by the conscious percept in the binocular rivalry condition and showed preference consistent with the physical switch condition ($p<0.05$, two-sided t-test), while 10% (34/348) of cells were significantly but inconsistently modulated. In AM, a face patch that receives input from ML (*Grimaldi et al., 2016*) and is the highest patch in the face patch hierarchy within IT (*Freiwald and Tsao, 2010*), the percentage of significant consistent modulation increased to 73% (153/210), with only 2% (5/210) showing significant inconsistent modulation. For both patches there was a clear correlation between modulation by the physical stimulus and modulation by the percept in binocular rivalry ($p = 2 \times 10^{-83}$, Pearson's $r = 0.70$, $N = 558$ cells). Thus, in a no-report paradigm, cells in IT exhibit modulations by the conscious percept that reflect their response selectivity to physically unambiguous inputs.

After eliminating the report confound, two important potential confounds remain. First, cells could be selective for the eye-of-origin of the fixation point that the animal is following (e.g., a cell could respond selectively to a fixation spot in the fovea of the left eye). Second, since we presented binocular stimuli using red/cyan anaglyph goggles, a confound could arise if cells were selective for the color of the fixation spot that is in the fovea. To control for these two potential confounds, we switched the colors and eye-of-origin of the face and object stimuli, that is, where the face and its corresponding fixation spot was previously presented in red in one eye, it was now presented in cyan in the other eye and vice versa for the object (*Figure 3—figure supplement 1*). If cells followed color or eye-of-origin, then all the dots in the upper right quadrant in *Figure 3—figure supplement 1a* should move to the lower left quadrant in *Figure 3—figure supplement 1b*. Instead, the majority of cells followed the object identity rather than color or eye-of-origin for both the physical and perceptual conditions ($p = 9 \times 10^{-42}$ for physical condition and $p = 10^{-19}$ for perceptual condition, one-sided t-test, $N = 313$ cells, alternative hypothesis: modulation indices for switched condition are greater than 0). This confirms that cells in IT cortex indeed represent the conscious percept rather than the color or eye-of-origin of the fixation spot.

The strong modulation by conscious percept in single cells suggests that we should be able to decode the percept on single trials from population activity. To test this, we performed recordings from multiple neurons simultaneously using S-probes with 32 electrode sites and passive Neuropixels-like probes with 128 electrode sites (see Materials and methods for details). *Figure 4* shows the recordings from face patch ML in one session using the Neuropixels probe. In this session, we recorded 81 cells simultaneously, of which 63 were face-selective (*Figure 4a*). An example population time course snippet of cells recorded simultaneously in the perceptual switch condition showed clearly stronger activity across the recorded population during perception of face compared to object (*Figure 4b*). The average population response across cells to perceptual switches is shown in *Figure 4c*. We found above chance decoding of the perceptual condition in all 12 sessions (in all but one session, responses were recorded in both ML and AM, and cells were pooled across the two

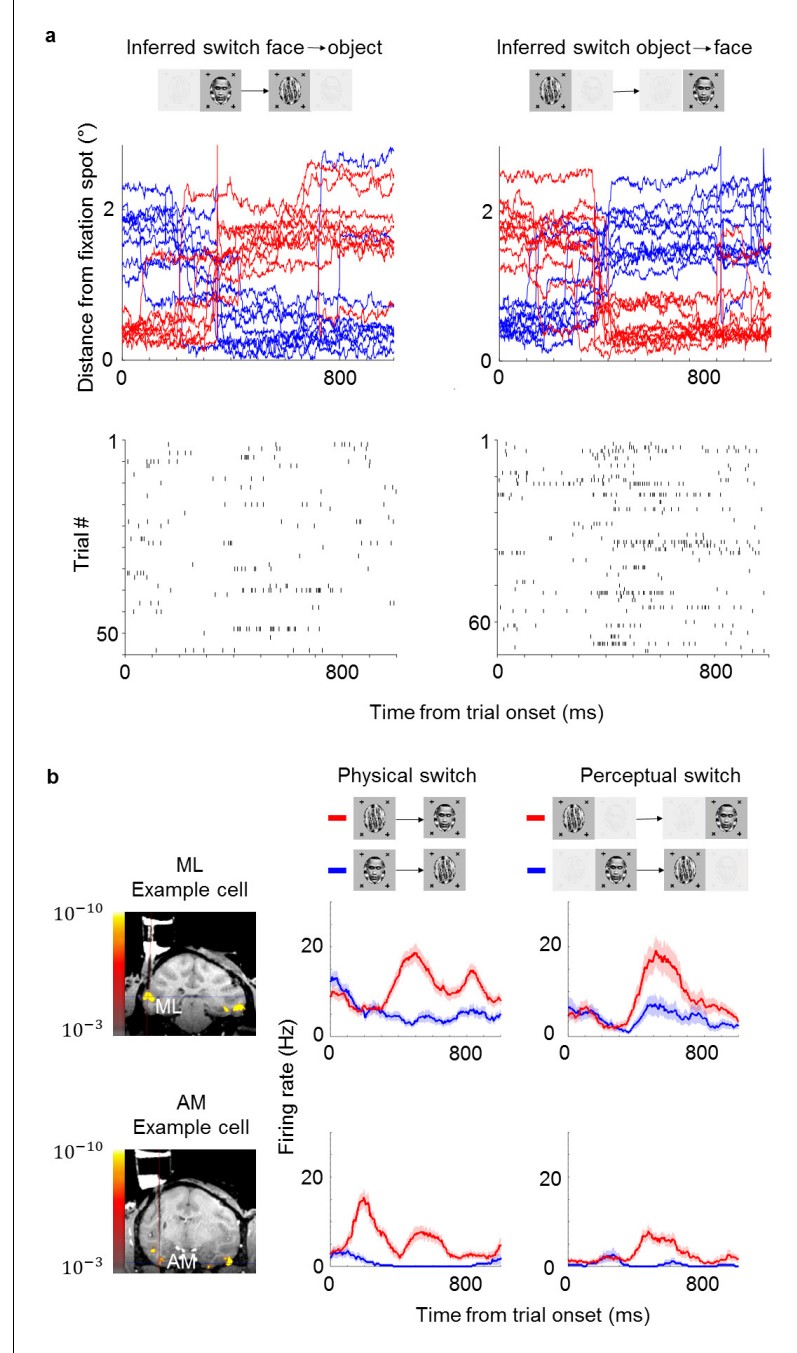

**Figure 2.** Example face cells modulated by both physical and perceptual switches. (a) Top: Example eye traces from a macaque performing the task aligned to a trial where the inferred percept switched from face to object (left) and from object to face (right). Red and blue curves indicate distances from the face and object fixation spots, respectively (as in *Figure 1b*). Bottom: Spike raster of an example ML cell recorded in the same session as for the top panel. Responses are aligned to all trials where the inferred percept switched from face to object (left) and from object to face (right). (b) Left: Coronal slices from magnetic resonance imaging scan showing recording locations for the two example cells in this figure (top: face patch ML, bottom: face patch AM). Color overlay shows functional magnetic resonance imaging activation to visually presented faces vs. non-face objects. Middle: Peristimulus histograms (PSTHs) show neuronal response time courses aligned to trial onsets where the visual stimulus was physically switched from face to object (blue) or from object to face (red). Right: PSTHs aligned to trial onsets where the inferred percept switched from face to object (blue) or object to face (red). ML cell is same cell as in (a). Shaded areas indicate standard error of the mean across trials.

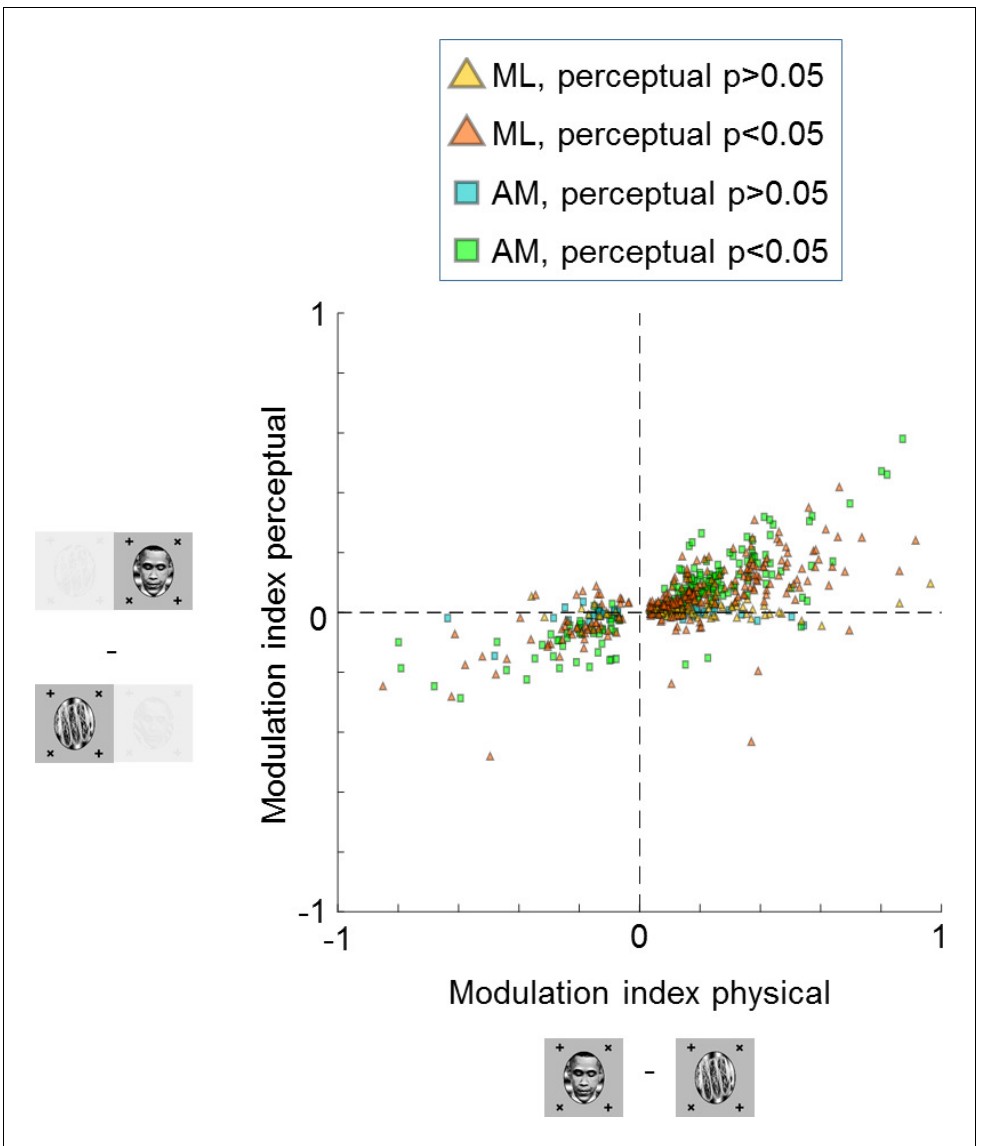

**Figure 3.** Large proportions of face cells show modulation by conscious percept. The scatterplot shows modulation indices $\left(R_{face} - R_{object}\right)/\left(R_{face} + R_{object}\right)$ measuring the difference in responses (i.e., average spike count $R$) on trials where the inferred percept was face vs. trials where the inferred percept was object for the physical monocular condition (x-axis) and perceptual binocular rivalry condition (y-axis). Yellow and orange triangles show cells from ML without and with significant difference between perceived face and perceived object response in the binocular rivalry condition, respectively. Blue and green squares show cells from AM without and with significant difference between perceived face and perceived object response in the binocular rivalry condition, respectively. The online version of this article includes the following figure supplement(s) for figure 3:

**Figure supplement 1.** Color and eye-of-origin confound control.

patches). Cross-validated accuracies of linear classifiers across different sessions are shown in *Figure 4d* (see Materials and methods). Decoding accuracy was 99% for the best session and 95% on average for the physical condition. For the perceptual condition, decoding accuracy was 88% on the best session and 78% on average.

Looking at the population time course, we noticed bursts of activity that appeared to be triggered by saccades, which occurred even when an object was perceived (blue epochs in *Figure 4b*; small black dots on top indicate detected saccades). This suggested to us that cells modulated by perception might still carry information about the physical stimulus: the bursts may have been caused by

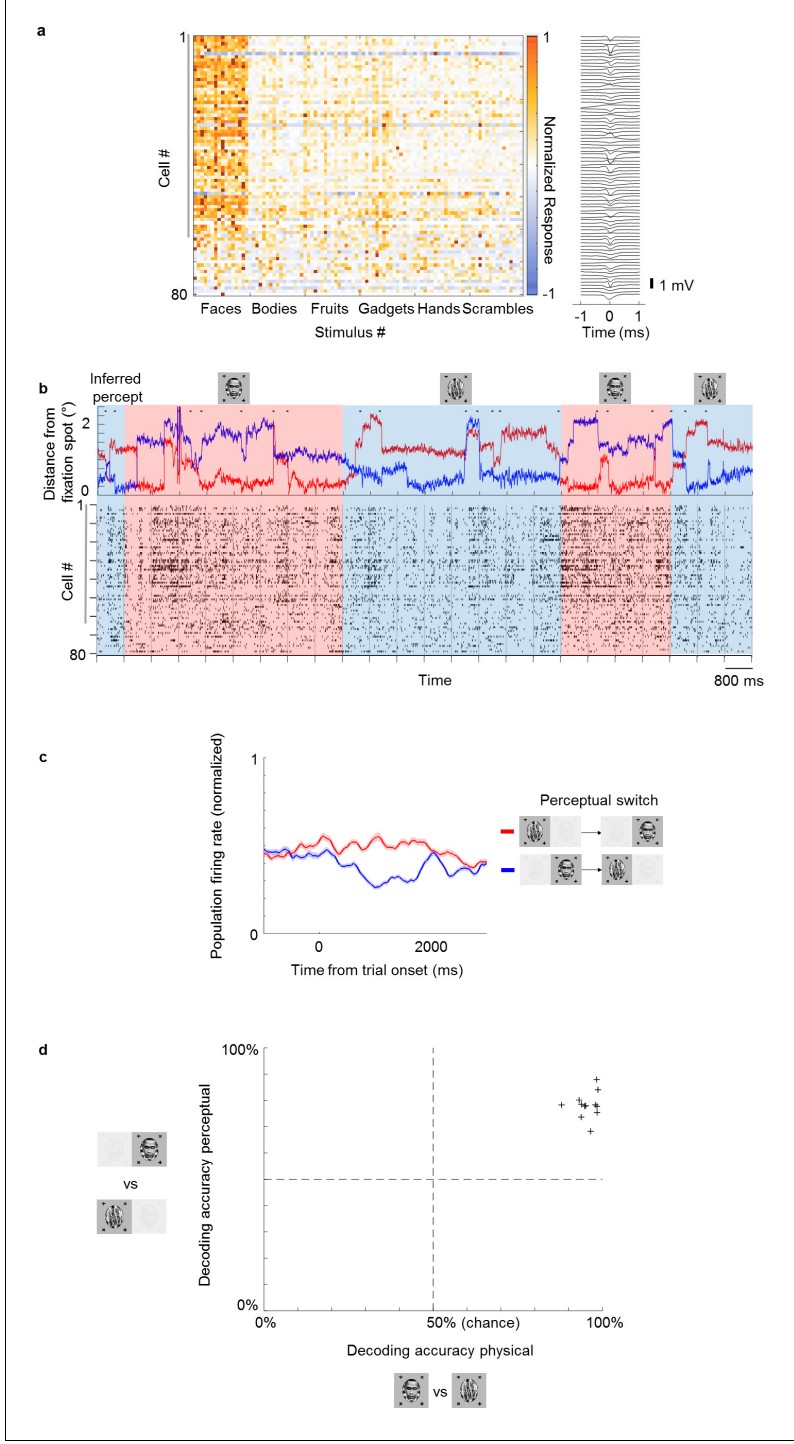

**Figure 4.** Multi-channel recordings allow decoding of conscious percept on single trials. (**a**) Left: Average responses (baseline-subtracted and normalized) of cells (rows) to 96 stimuli (columns) from six categories, including faces and other objects. Right: Waveforms of cells corresponding to rows on the left. Gray vertical bar on left indicates cells that significantly preferred face over object in the physical condition (p<0.05). (**b**) Top: Example eye trace across 24 trials as in *Figure 1b* during binocular rivalry (i.e., only perceptual, no physical switches). The inferred percept across trials according to eye trace is indicated by shading (red = face, blue = non face object). Small black dots on top of eye traces indicate time points where our method detected saccades (see Materials and methods), which are used in *Figure 5*. Bottom: Response time course snippet of a population of 81 neurons recorded with a Neuropixels-like probe in ML simultaneously to the eye trace at top. Each row represents

*Figure 4 continued on next page*

*Figure 4 continued*

one cell; ordering same as in (a). Face-selective cells indicated by gray vertical bar on left. (**c**) Normalized average population response across all significantly face-selective ML cells recorded from one Neuropixels session (same session as in a and b) to perceptual switch from object to face (red) and face to object (blue). Shaded areas indicate standard error mean across cells. (**d**) Cross-validated decoding accuracy of a linear classifier trained to discriminate trials of inferred percept face vs. inferred percept object for the physical switch condition (x-axis) and perceptual switch condition (y-axis). Each plus symbol represents a session of neurons recorded simultaneously with multi-channel electrodes.

responses to the suppressed face stimulus. To investigate this further, we selected cells that (i) showed both significant physical and perceptual modulation and (ii) consistently preferred the face over the object. We then averaged responses across these cells and computed response time courses triggered by individual saccades, grouped by whether a saccade occurred during a trial inferred to be face or object, respectively (*Figure 5*). We observed response modulations for both physical and perceptual conditions starting around 130 ms after saccade onset (*Figure 5a*). In the physical condition, a saccade during an object epoch led to response suppression, while a saccade during a face epoch led to response increase. In striking contrast, in the rivalry condition saccades led to response increase in both object and face epochs. As a consequence, during rivalry the response difference to a saccade between face and object, though significant ($p = 6 \times 10^{-23}$, two-sample t-test, $N = 701$ saccades for object, $N = 703$ saccades for face), was weaker than during the physical condition. Computing histograms of responses averaged across neurons for individual saccades shows that responses in the rivalry condition were less bimodal and spanned a smaller range compared to the physical condition (*Figure 5b*). Importantly, this difference in response profiles between physical and perceptual conditions was apparent even when pooling across both face and object trials (*Figure 5b*, middle), and *hence cannot be explained by mistakes in inferring the percept from eye movements*. We computed the absolute value of these responses and found the difference in response distributions to be significant (*Figure 5b*, right, $p = 6 \times 10^{-35}$, two-sample t-test on absolute value distributions, $N = 229$ saccades for physical condition, $N = 1404$ saccades for perceptual condition).

The observation of different response profiles for physical and perceptual conditions was not specific for saccades: histograms were also less bimodal and spanned a smaller range for the rivalry condition when triggering responses on trial onsets rather than saccades in both ML (*Figure 5—figure supplement 1a*, $p = 9 \times 10^{-15}$, two-sample t-test on absolute value distributions, $N = 150$ trials for physical condition, $N = 571$ trials for perceptual condition) and AM (*Figure 5—figure supplement 1b*, $p = 0.0014$, two-sample t-test on absolute value distributions, $N = 120$ trials for physical condition, $N = 480$ trials for perceptual condition). Therefore, it appears that throughout rivalry, for perceptually modulated cells, response differences to face and object are less pronounced than in the physical condition, and this is true in both ML and AM. One tantalizing explanation for this phenomenon is that perceptually modulated cells may be multiplexing information about both the physical stimulus and the perceptual state during single trials, allowing both to be simultaneously represented across the face patch hierarchy.

Is it possible that the apparent responses to the suppressed face were due to incomplete suppression, leading to piecemeal percepts on some trials? We performed simulations of the worst-case effect of mixture, in which the percept would be exactly half-face and half-object, by taking the responses of the physical condition and averaging responses to face and body on a specific proportion of trials. The simulated distributions only became statistically indistinguishable from the observed binocular rivalry condition if 50–70% of trials were mixed percepts of half-face and half-body. This is markedly inconsistent with reports from every human subject that on most trials they did not perceive any mixture. We of course cannot be absolutely sure that monkeys do not experience mixed percepts significantly more often than humans. Yet, under the reasonable assumption that percepts were similar in the two species, trials with mixed or piecemeal percepts cannot account for the difference in response distributions between physical and perceptual conditions.

To directly test the hypothesis that cells multiplex information about the perceptually dominant and suppressed stimulus, we performed a new experiment in which we varied the identity of the suppressed stimulus. In this experiment, instead of having only two rivalling stimuli, we used three images A, B, and C to create two different binocular rivalry stimuli, (A,B) and (A,C), presented in

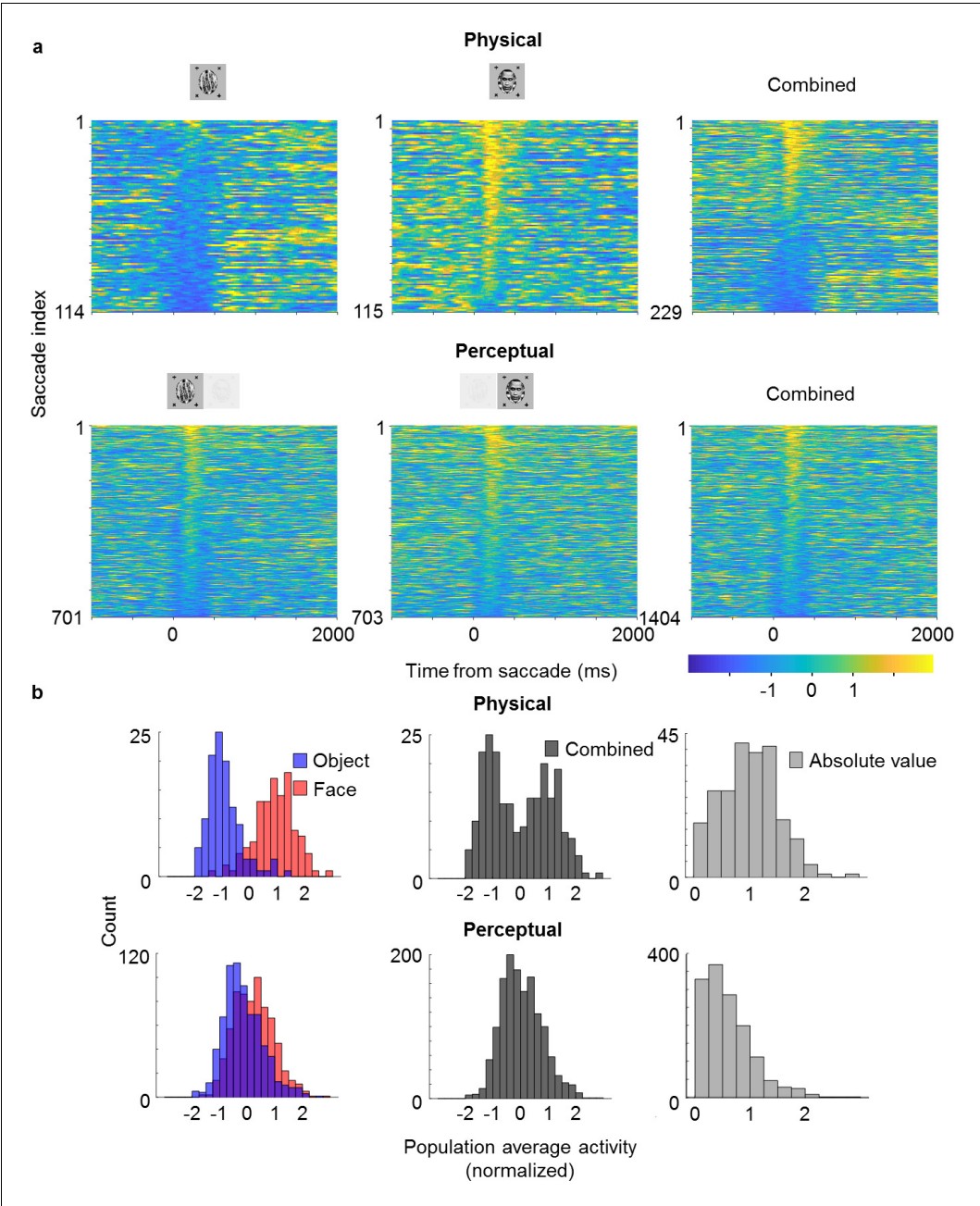

**Figure 5.** Saccade-triggered responses are less bimodal during rivalry. (a) Single-trial responses during saccades averaged across simultaneously recorded ML neurons from the same session as in *Figure 4b* that were significantly face-selective for both physical and perceptual conditions. Individual neuron responses were normalized to make the mean object response −1 and the mean face response +1. Rows of each plot correspond to response time courses to individual saccades, aligned to saccade onset, and sorted by average response during 0–400 ms after saccade onset. Top: Physical condition. Bottom: Perceptual condition. Left, middle, and right columns correspond to saccades during object epochs, face epochs, and across both, respectively. The difference between perceptual and physical conditions in the third column shows that this difference cannot be simply attributed to mislabeling of perceptual state by the no-report paradigm. (b) Histograms of saccade-aligned responses averaged across a time window of 0–400 ms after saccade onset and across neurons (after normalizing as in (a)) that were significantly modulated for both physical and perceptual conditions. Top: Physical condition. Bottom: Perceptual condition. Left: Saccades for face and object plotted separately in red and blue, respectively. Middle: Saccades for either face or object epochs plotted in gray. Right: Absolute values of normalized responses plotted in light gray.
The online version of this article includes the following figure supplement(s) for figure 5:

**Figure supplement 1.** Lack of bimodality is a general trademark of rivalry.

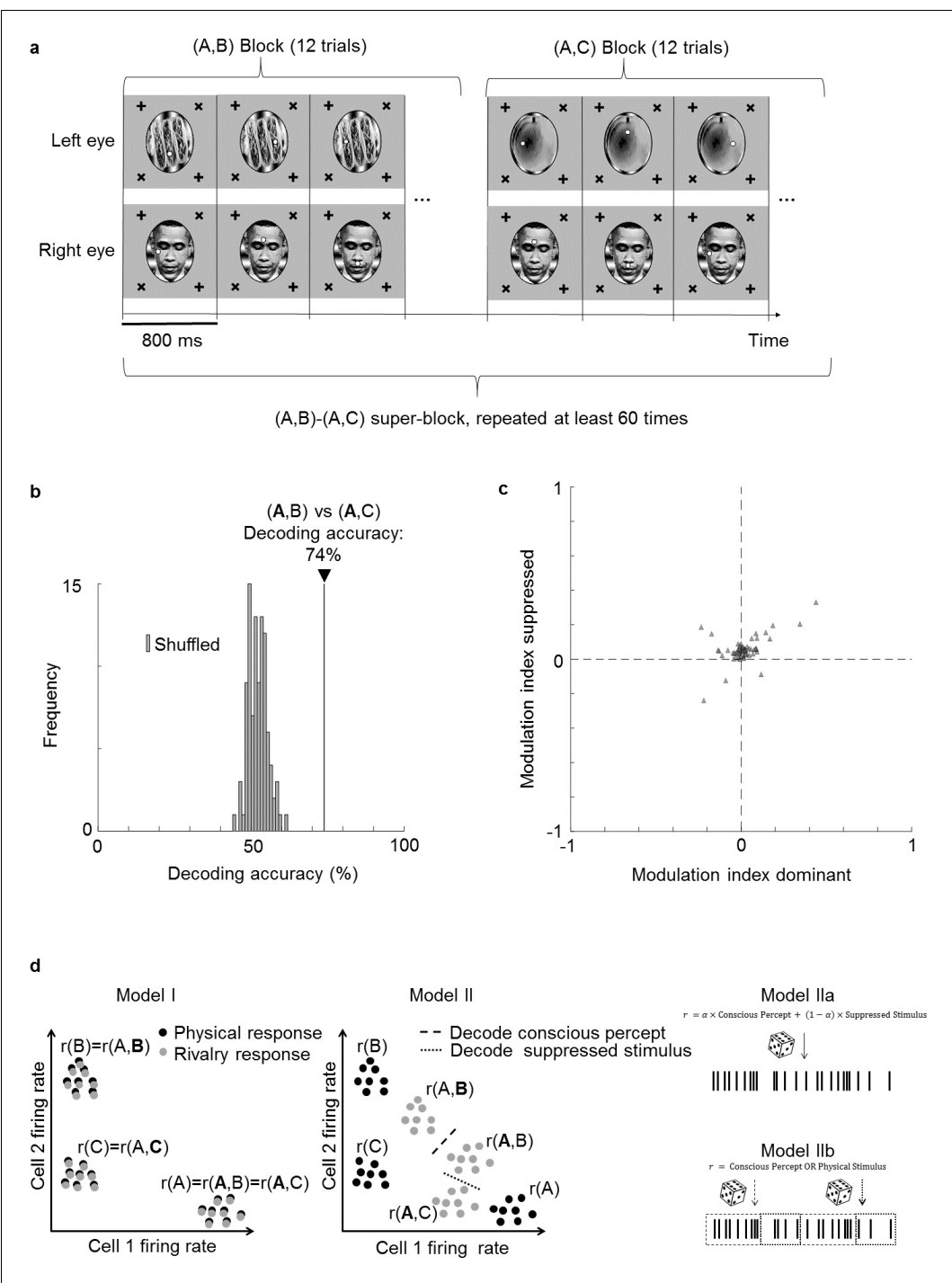

**Figure 6.** Face cells multiplex information about both the perceptually dominant and perceptually suppressed stimulus. (**a**) Schematic of experiment design. Two types of binocular rivalry stimuli consisting of image pairs (A,B) and (A,C), respectively, were presented. During one image pair block, 12 trials corresponding to the 12 positions of the two fixation spots were presented in randomized order before the next block corresponding to the other image pair was presented. This was repeated more than 60 times. (**b**) Decoding accuracy for distinguishing (A,B) from (A,C) was 74% (black vertical line), even though the conscious percept was A for both trial types. As a control, we shuffled labels 100 times and attempted to perform decoding. Gray bars show the distribution of decoding accuracies for these 100 shuffle iterations. (**c**) Scatterplot showing dominant stimulus modulation indices $MI_{dominant} = (R_{AB} - R_{AC})/(R_{AB} + R_{AC})$ on the x-axis and suppressed stimulus modulation indices $MI_{suppressed} = (R_{AB} - R_{AC})/(R_{AB} + R_{AC})$ on the y-axis. Each triangle represents 1 of 66 physically selective cells recorded in one session from face patch ML with a 64-ch S-probe. (**d**) Schematic of three possible models for how perceptually modulated neurons may encode consciously perceived and suppressed stimuli during binocular rivalry. Left: (I) Neural responses encode the conscious percept in binocular rivalry identically to the corresponding unambiguous physical stimulus; x and y axes

*Figure 6 continued on next page*

*Figure 6 continued*

represent two dimensions of neural state space. Middle: (II) Responses during binocular rivalry lie in between the two stimuli but are biased toward the dominant stimulus. Right: (IIa) Spikes reflect a weighted sum of consciously perceived and suppressed stimuli and are generated through a Poisson process based on average firing rates. (IIb) Two different types of spikes, defined, for example through a temporal code, encode the consciously perceived and veridical physical stimulus, respectively. The time course in this schematic is from a single perceptual dominance period and divided into different epochs that represent either the conscious or physical stimulus.

separate blocks (*Figure 6a*). This allowed us to keep the dominant percept fixed as image A, and compare responses to trial types (**A**,B) and (**A**,C) to test whether neural responses could discriminate the suppressed stimulus (bold font indicates the dominant image, as inferred by eye movements). We trained a linear decoder to distinguish between trial types (**A**,B) and (**A**,C). Remarkably, the decoding accuracy for distinguishing the two trial types was 74% (*Figure 6b*). For comparison, the decoding accuracy for distinguishing (A,**B**) vs. (A,**C**) from the same cell population was 88%. Thus, while the conscious percept can be decoded better than the suppressed stimulus, face cells do encode significant information about the latter. Potential mislabeling of trials by the no-report paradigm could not account for this decoding accuracy (see Supplementary text).

Do the same cells multiplex information about both the conscious and suppressed stimuli, or are there two distinct subpopulations, one encoding the conscious stimulus and another encoding the suppressed stimulus? To address this question, we compared modulation indices for the dominant stimulus with modulation indices for the suppressed stimulus for each cell. For the former, we fixed the suppressed stimulus while varying the dominant stimulus, that is, we compared responses to trial type (A,**B**) with responses to trial type (A,**C**) to compute the modulation index $MI_{dominant} = (R_{A\mathbf{B}} - R_{A\mathbf{C}})/(R_{A\mathbf{B}} + R_{A\mathbf{C}})$. For the latter, we fixed the dominant stimulus while varying the suppressed stimulus, that is, we compared responses to trial type (**A**,B) with responses to trial type (**A**,C) to compute the modulation index $MI_{suppressed} = (R_{\mathbf{A}B} - R_{\mathbf{A}C})/(R_{\mathbf{A}B} + R_{\mathbf{A}C})$ (*Figure 6c*). We found a positive correlation between dominant stimulus modulation indices and suppressed stimulus modulation indices ($p = 1.4 \times 10^{-6}$, Pearson's $r = 0.55$, $N = 66$ physically selective cells). This suggests that cells strongly modulated by the dominant stimulus tend to be similarly modulated by the suppressed stimulus. Thus there are not two separate populations of cells that encode conscious and unconscious stimuli.

In summary, our findings indicate that the neural correlate of consciousness in IT does not reside in a subset of cells perfectly reflecting consciousness but rather in a population code. This is supported by the findings that (i) modulation by the conscious percept is weaker than modulation by the physical stimulus (*Figures 3* and *5*), (ii) both consciously perceived and suppressed stimuli can be decoded from the same population (*Figure 6b*), and (iii) modulation indices for consciously perceived and suppressed stimuli are correlated in single cells (*Figure 6c*).

## Discussion

In this study, we developed a new no-report paradigm for tracking conscious state and used it to investigate the neural correlate of consciousness in face patches within macaque IT cortex. We made two new findings. First, we found that face patches ML and AM are modulated by conscious perception and do not merely encode the physical input. Importantly, monkeys in this study had never been trained to actively report their percept. Instead, we were able to infer their percept from eye movements using a new no-report paradigm. Thus activity modulations attributed to switches in conscious perception in IT cannot be explained simply by active report. Second, we found that cells in face patches are modulated by the identity of both the consciously perceived stimulus and the suppressed stimulus, such that both stimuli can be read out from the same population using different linear decoders. This finding challenges the widely held notion that in IT cortex almost all neurons respond only to the consciously perceived stimulus.

Previous single-unit recordings in IT cortex using active report to infer the percept found 90% of cells represent the conscious percept (*Sheinberg and Logothetis, 1997*). Here, we found proportions of 57% in ML and 73% in the more anterior patch AM. The quantitative difference may be due to several factors including different recording sites (Sheinberg and Logothetis recorded from both upper and lower banks of the superior temporal sulcus in a less specifically targeted manner),

imperfect accuracy of the no-report paradigm, and differences in stimuli and analysis methods. Importantly, our results confirm that the majority of cells in IT cortex do represent conscious perception. Furthermore, this new paradigm makes studies of consciousness in monkeys more accessible, by replacing the need to train the animal to signal its conscious percept (which can be a laborious process) with a simple task that only requires animals to follow a fixation spot.

Our results show that for cells that are modulated by conscious perception, the modulation is not 'all-or-none'. Instead, the average response modulation during the perceptual condition was weaker than during the physical condition (*Figure 3*). This was also observed in a previous study of rivalry (*Sheinberg and Logothetis, 1997*), but somehow, this fact has been forgotten in popular lore surrounding the neural correlates of consciousness. For example, the Wikipedia entry for 'neural correlates of consciousness' states that "in [inferior temporal cortex] almost all neurons responded only to the perceptually dominant stimulus, so that a 'face' cell only fired when the animal indicated that it saw the face and not the pattern presented to the other eye". We think the reason this fact - the decreased average modulation of IT cells by switches in conscious percept compared to switches in physical stimulus - has not garnered much attention up to now is that it could, at least up to now, be simply explained by imperfect labeling of the animal's perceptual state.

The key question is: *what happens during single trials?* In the rivalry condition, do responses in single trials look like those to either physically presented faces or objects? By recording from a large number of face cells simultaneously using a novel 128-electrode site probe specifically designed for use in primates, we could address this question for the first time. Surprisingly, we found a dramatically different response profile on single trials between the perceptual and physical conditions (*Figure 5*). Although in the physical condition responses clustered into two groups, in the rivalry condition responses appeared unimodal, lying in between the two clusters for the physical condition. This suggests that single cells are multiplexing the conscious percept and the veridical physical stimulus during single trials. To directly test this hypothesis, we presented more than one binocular rivalry stimulus, created from pairs of three images, and found that the subconscious stimulus could indeed by decoded from face patch activity. Moreover, the same cells that were strongly modulated by the conscious percept also tended to be strongly modulated by the suppressed stimulus, ruling out the existence of a subpopulation of cells in IT purely reflecting consciousness. These findings strongly suggest that rivalry is not fully resolved before IT. It remains an open question where and how the conscious percept is ultimately isolated from the suppressed stimulus to produce conscious awareness of the former and not the latter.

In *Figure 6d*, we sketch three models for how perceptually modulated cells in IT cortex could encode stimuli during binocular rivalry. In Model I, cells exactly reflect the conscious percept, encoding it the same way they would encode an unambiguous stimulus. In Model II, the response during binocular rivalry is in between the responses to the two unambiguous stimuli, with the contributions of the two stimuli weighted differently depending on which stimulus is dominant and which stimulus is suppressed. Thus, both the consciously perceived stimulus and suppressed stimulus can be decoded using two different decoders. For Model II, one can further distinguish between two different sub-models depending on whether consciously perceived and suppressed stimuli are encoded by different subsets of spikes or not: in Model IIa, spikes are stochastically generated from the average firing rate on a trial, which is determined by a linearly weighted sum of consciously perceived and suppressed stimuli. Alternatively, in Model IIb, there are two different types of spikes that encode the conscious percept or physical stimulus, respectively. The type of a spike may depend on the phase of a high-frequency oscillation at which the spike occurs (the oscillation would need to be faster than alternations in perceptual dominance), or on whether the spike occurs synchronously with spikes from other neurons. Unlike Model IIa, Model IIb harbors an explicit neural correlate of the conscious percept within a subset of spikes. Importantly, our result that the suppressed stimulus can be decoded rules out the cartoon picture of Model I. Our findings are compatible with both Models IIa and IIb, and future experiments may be able to distinguish between the two.

Compared to previous approaches that attempted to isolate representations of the conscious percept, our new no-report binocular rivalry paradigm has several advantages. For flash suppression, where a stimulus flashed in one eye suppresses the stimulus in the other eye, report is also not required (*Tsuchiya and Koch, 2005*; *Wilke et al., 2003*; *Wolfe, 1984*). However, in that case, the physical input when the target is perceived is not identical to that when it is suppressed, and thus any modulation observed may be driven entirely externally. Indeed, it is known that if a distractor

stimulus is presented simultaneously with a preferred stimulus, the response can be reduced compared to when the preferred stimulus is presented alone, due to simple normalization mechanisms (*Bao and Tsao, 2018*). Another paradigm that has been widely used to study the neural correlates of consciousness is backward masking. Here, the stimulus is presented for such a short time before being masked that sometimes it enters consciousness and sometimes not (*Breitmeyer et al., 1984*). So far, backward masking has always relied on report. Also, it is more susceptible to modulations arising from bottom-up withdrawal of attention or low-level (e.g., retinal) noise, whereas in binocular rivalry perceptual switches appear to be internally generated. One potential confound described by Block as the 'bored monkey problem' is that the monkey may still be thinking about whether it is perceiving object or face and internally report it even if it is not required to actively report it (*Block, 2020*). It is methodologically very difficult to entirely remove this confound, but the fact that monkeys had to simultaneously perform a very challenging unrelated task of saccading to jumping fixation points should at least alleviate this concern.

Alternative approaches to the no-report paradigms of *Frässle et al., 2014* have been developed in which the monkey or human subject is unaware of when a perceptual switch is happening and hence cannot report it, either due to anesthesia or due to the difference in stimuli being too subtle to report. *Brascamp et al., 2015* reported that fMRI responses to binocular rivalry switches in fronto-parietal regions disappear when the difference between the percepts is made so subtle that subjects cannot report it; however, it is possible that the difference between the percepts was just too small to be picked up by the fMRI signal. *Zou et al., 2016* created rivalry stimuli from orthogonal gratings where the grating in one eye was flickered fast enough that it was perceived as uniform gray and only produced fMRI activations in early visual cortex. These stimuli evoked rivalry according to behavioral reports whereas physically uniform stimuli do not, indicating that competition occurred in early visual cortex. In another study consistent with competition in early visual cortex, *Xu et al., 2016* performed optical imaging in V1 while monkeys were anesthetized. They found that during binocular rivalry activations clearly alternated in counter-phase between left eye and right eye dominance columns. We note that competition occurring in V1 is not incompatible with our findings, although our findings suggest that rivalry is unlikely to be fully resolved in early areas, given our ability to decode the suppressed stimulus from cells in IT. It should also be noted as a caveat that hemodynamic signals, as measured in the above studies by fMRI or optical imaging, only indirectly reflect neural activity and have previously shown discrepancies with single-unit responses (*Leopold and Logothetis, 1996*; *Tong and Engel, 2001*). Overall, to the best of our knowledge, the current study describes the representation of conscious and subconscious stimuli in IT cells in the most confound-free way to date. Our study complements a study conducted in parallel by *Kapoor et al., 2020* that found modulations by conscious percept in prefrontal cortex using a different no-report paradigm based on optokinetic nystagmus.

The existence of two directly connected functional modules with a hierarchical relationship (ML and AM) that both encode the conscious percept of a particular type of object opens the possibility for future studies to investigate how changes in conscious percept are coordinated across the brain. Recordings and perturbations in multiple face patches simultaneously using high-channel count recordings may reveal whether switches occur in a feedforward or feedback wave, and thus yield insight into how our interpretation of the world can be rendered consistent across different levels of representation.

## Materials and methods

All animal procedures in this study complied with local and National Institute of Health guidelines including the US National Institutes of Health Guide for Care and Use of Laboratory Animals. All experiments were performed with the approval of the Caltech Institutional Animal Care and Use Committee (IACUC). The behavioral experiment with human subjects for the human psychophysics experiment complied with a protocol approved by the Caltech Institutional Review Board (IRB).

### Targeting

Two male rhesus macaques were implanted with head posts and trained to fixate on a dot for juice reward. We targeted face patches ML and AM in IT cortex for electrophysiological recordings. ML and AM were identified using fMRI. Monkeys were scanned in a 3T scanner (Siemens), as described

previously (*Tsao et al., 2006*). MION contrast agent was injected to increase signal-to-noise ratio. During fMRI, monkeys passively viewed blocks of faces and blocks of other objects to identify face-selective patches in the brain. Recording chambers (Crist) were implanted over ML and AM. Guide tubes were inserted into the brain 4 mm past the dura through custom-printed grids placed inside the chamber, and electrodes were advanced to the target through the guide tube. Both chamber placement and grid design were planned with the software Planner (*Ohayon and Tsao, 2012*). After insertion of tungsten electrodes, correct targeting of the desired location was confirmed with anatomical MRI scans.

## Electrophysiology

Recordings were performed using tungsten electrodes (FHC) with 1 MΩ impedance and, after correct targeting was confirmed, with 32-channel S-probes (Plexon) with 75 μm and 100 μm inter-electrode distance, and, in three sessions, with passive Neuropixels-like probe prototypes (IMEC) (*Dutta et al., 2019*; *Jun et al., 2017*; *Trautmann et al., 2019*). These prototypes were a limited stock of test devices that were developed and used for testing as part of the development of primate Neuropixels probes and are not available for other labs. Unlike the final product, the prototypes had 128 passive electrode sites across 2 mm (arranged in two parallel staggered bands), but used the same electrode materials and shank specifications (45 mm total shank length). In the additional experiment performed to decode the suppressed stimulus (*Figure 6*), we recorded with a novel 64-ch. S-probe in face patch ML. All electrodes were advanced to the target using an oil hydraulic Microdrive (Narishige). Neural signals were recorded using an Omniplex system (Plexon). Local field potentials were low-pass filtered at 200 Hz and recorded at 1000 Hz, and units were high-pass filtered at 300 Hz and recorded at 40 kHz. Only well-isolated units were considered for further analysis.

## Task

Monkeys were head fixed and viewed an LCD screen (Acer) of 47° size in a dark room. Monkeys viewed stimuli of 5° size wearing red/cyan anaglyph goggles custom made with filters to match the red and green/blue emission spectrum of the screen, respectively, so that inputs to left and right eyes could be controlled independently. Emission spectra were measured using a PR-650 SpectraScan colorimeter (Photo Research). Eye position was monitored using an infrared eye tracking system (ISCAN). The camera recorded one eye through the red/cyan anaglyph filter. We measured the precision of ISCAN eye positions by computing the absolute value of distances between 1 ms adjacent eye data. The median and 99% confidence interval of this jitter was 0.038° and 0.34°, respectively. Note that these confidence intervals should not be contaminated by saccades which occur less frequently than 10 Hz and therefore make up less than 1% of the distribution. In the first phase of the experiment, monkeys passively viewed at least five repeats of 61 screening stimuli in pseudorandom order (250 ms ON time, 100 ms OFF time) with a fixation spot of 0.25° diameter in the center of the screen. Screening stimuli consisted of 20 images of faces and 41 images of non-face objects. During this phase, monkeys received a juice reward for maintaining fixation for at least 3 s. Subsequently, for the main experiment, stimuli contained one or two fixation spots at one of four possible locations (top, bottom, left, and right, 1° from the center) and were presented for 800 ms ON time and 0 ms OFF time. In the case of two fixation spots, stimuli contained one fixation spot per eye and the two spots never appeared at the same location. During this phase, the monkey received a juice reward if the monkey maintained fixation within 0.5° of one of the fixation spots for at least half of the trial duration (i.e., 400 ms, not required to be contiguous). Stimuli during the main experiment included (1) a monocular face/monocular object with one fixation spot and (2) a binocular stimulus composed of a face and a fixation spot in one eye, and an object and a second fixation spot in the other eye. During the binocular rivalry condition, even though the same stimulus was presented continuously, we refer to the 800 ms duration, after which the two fixation spots would change position, as one trial. To improve rivalry and minimize periods of mixture, face and object stimuli were presented at high contrast on backgrounds consisting of gratings that were orthogonal in the two eyes. Moreover, we applied orthogonal orientation filters (with concentration $\sigma_{angle} = 0.5°$) to the face and object stimuli, respectively, to increase local orientation contrast and further reduce periods of mixture. For human subjects, stimuli were identical except that the trial duration was 2000 ms, since

they had not been extensively trained on the task unlike monkeys and hence needed more time to saccade to the jumping fixation spots. During the additional session performed for decoding the suppressed stimulus (*Figure 6*), we presented stimuli in a block design. Each block corresponded to an image pair, for example (A,B), where each fixation position was presented in randomized order, that is, eight trials for the physical condition (including four trials of unambiguous A and four trials of unambiguous B), and 12 trials for the perceptual condition, after which another block was presented (*Figure 6a*). We repeated this design so that each image-pair block was presented for at least 60 repetitions.

### Online analysis

Spikes were isolated and sorted online using the PlexControl software (Plexon). During the screening phase, the average number of spikes during the time window from 100 ms to 300 ms was calculated for each unit and stimulus. For each stimulus, the average response across units was determined after normalizing the response of each unit by subtracting the mean and dividing by the standard deviation for the unit. Subsequently, the face stimulus with the highest average response and the object stimulus with the lowest average response were chosen to generate stimuli for the main experiment.

### Offline analysis

For human subjects, the inferred percept based on button-presses on a given trial was determined according to the last report the subject made before the end of the trial. For humans and monkeys, we also determined their inferred percept based on eye movements depending on which fixation spot they fixated on if they fixated on one of the fixation spots for at least half of the trial duration (i.e., 400 ms for monkeys or 1000 ms for humans, not required to be contiguous). We computed L1 norms for the distance between eye position and a given fixation spot (*Figures 1b*, *2a*, and *4b*). We accounted for an average saccade delay of 350 ms, by analyzing the eye data from 350 ms after trial onset until 350 ms after trial end. For *Figure 3*, *Figure 3—figure supplement 1*, *Figure 4d*, *Figure 5—figure supplement 1*, and *Figure 6* in order to exclude trials during which the percept switched back to the opposite percept, we also required the following trial to have the same inferred percept as the current trial. Spikes were re-sorted using the software OfflineSorter (Plexon). For the Neuropixels prototypes, since the high density of electrodes allowed the same neuron to appear on multiple channels, we used Kilosort2 to re-sort spikes (*Pachitariu et al., 2016*). A total of 653 and 481 cells were recorded in monkey A and monkey O, respectively. To correct for delays in stimulus presentation, we used a photodiode that detected the onset and offset of the stimuli. The output of the photodiode was fed into the recording system and later used to synchronize the onset of the stimulus and the neurophysiological data during offline analysis. Peristimulus time histograms (PSTHs) were smoothed with a box kernel (100 ms width). For computing modulation indices we used the average spike count across trials as response. Decoding analysis was performed with a support vector machine with a linear kernel (Matlab fitcsvm) trained to discriminate trials where the inferred percept was face or object, respectively. As predictor variables we used the spike count during the 800 ms of each trial for all simultaneously recorded neurons. All decoding accuracies were cross-validated. In more detail, one trial was chosen for testing and the remaining trials for training; this was repeated for all trials to compute decoding accuracies. Criteria for detecting a saccade were as follows: A saccade was detected at time t if the distance between the mean eye position during t−100,...t−2 ms and the mean eye position during t+2,...t+100 ms was greater than 0.5°, and the eye position during t−100,...t−2 ms and t+2,...t+100 ms, respectively, stayed within 0.5° of the respective mean for at least 80% of the duration of each period. We also required consecutive saccades to be at least 100 ms apart from each other. All analyses were performed using Matlab (MathWorks).

## Acknowledgements

This work was supported by HHMI and the Simons Foundation. We are grateful to members of the Tsao lab for feedback on the manuscript, Varun Wadia for helping us collect the human subject data, Audo Flores for animal support, Daniel Wagenaar and Eric Trautmann for technical assistance,

and Barun Dutta, Tim Harris, Tirin Moore, Michael Shadlen, Krishna Shenoy, and HHMI for contributions to development of NHP Neuropixels probes.

## Additional information

### Funding

| Funder | Author |
| --- | --- |
| Howard Hughes Medical Institute | Doris Y Tsao |
| Simons Foundation | Doris Y Tsao |

The funders had no role in study design, data collection and interpretation, or the decision to submit the work for publication.

### Author contributions

Janis Karan Hesse, Conceptualization, Data curation, Software, Formal analysis, Validation, Investigation, Visualization, Methodology, Writing - original draft, Writing - review and editing; Doris Y Tsao, Conceptualization, Resources, Supervision, Funding acquisition, Methodology, Writing - original draft, Project administration, Writing - review and editing

### Author ORCIDs

Janis Karan Hesse (iD) https://orcid.org/0000-0003-0405-8632
Doris Y Tsao (iD) https://orcid.org/0000-0003-1083-1919

### Ethics

Human subjects: The behavioral experiment with human subjects for the human psychophysics experiment complied with a protocol approved by the Caltech Institutional Review Board (IRB 19-0903). Informed consent was obtained from all subjects.

Animal experimentation: All animal procedures in this study complied with local and National Institute of Health guidelines including the US National Institutes of Health Guide for Care and Use of Laboratory Animals. All experiments were performed with the approval of the Caltech Institutional Animal Care and Use Committee (IACUC), under protocol #1574.

### Decision letter and Author response

Decision letter https://doi.org/10.7554/eLife.58360.sa1
Author response https://doi.org/10.7554/eLife.58360.sa2

## Additional files

### Supplementary files

• Transparent reporting form

### Data availability

All data generated or analysed during this study are included in the manuscript and supporting files.

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

## Appendix 1

### Supplementary text

In the Results section we found that the suppressed stimulus, that is, B or C in binocular rivalry trials (**A**,B) vs. (**A**,C), where **A** is the consciously perceived image, could be decoded from neural activity with 74% accuracy. A natural question arising from the decoding accuracy of 74% is whether this could have been due to mislabeling by the no-report paradigm. On some trials, the conscious percept may have been mislabeled as (**A**,B) or (**A**,C) and actually have been (A,**B**) or (A,**C**), respectively. In this case, even if cells only encode the conscious percept and not the suppressed stimulus, the decoding accuracy would be higher than chance, as on those mislabeled trials, the decoder could successfully discriminate based on a difference in conscious percept. To address this concern, below we estimate the worst-case decoding accuracy increase we could expect from mislabelings under the null hypothesis that neurons do not encode the suppressed stimulus. For image pair (A,B), we could decode (**A**,B) vs. (A,**B**), that is, whether A or B was consciously perceived as in *Figure 4d*, with 89% accuracy in this session. If we had recorded more neurons, or neurons that were more selective, we would expect a decoding accuracy at least as high. Since there is physically no difference between trial types (**A**,B) and (A,**B**), any information that the decoder was able to acquire must have come from the difference in conscious percept. Thus, we can use 89% as a lower bound for the estimated accuracy of the no-report paradigm in inferring the correct conscious percept in this session. Under the null hypothesis that neurons only encode the conscious percept, the decoding accuracy for distinguishing (**A**,B) from (**A**,C) for 89% of trials should be chance. For the remaining 11% of trials, the conscious percept may have been B or C, respectively. Even if the decoder can decode all of these mislabeled trials with 100% accuracy (which is an overestimate), the decoding accuracy across all trials would be at most $89\% \times 50\% + 11\% \times 100\% = 55.5\%$. So even in the worst-case, the mislabeled trials would not lead to the observed decoding accuracy of 74%. This suggests that face cells do indeed encode the suppressed image.

