## [Decision Letter]

**Acceptance summary:**

Binocular rivalry is a prominent type of bistable perception (illusion), in which observer's conscious perception automatically switches while stimuli remain unchanged. The present study combines cutting-edge neurophysiological recordings and a novel no-report paradigm to revisit whether macaque inferotemporal (IT) cortex correlates with the animal's conscious percept. The results are provocative, suggesting that a) cells in the IT cortex are modulated by conscious percept; b) single cells may multiplex representation of illusory percept and physical stimulus.

**Decision letter after peer review:**

Thank you for submitting your article "Representation of conscious percept without report in the macaque face patch network" for consideration by *eLife*. Your article has been reviewed by three peer reviewers, and the evaluation has been overseen by a Reviewing Editor and Floris de Lange as the Senior Editor. The following individual involved in review of your submission has agreed to reveal their identity: Brad Duchaine (Reviewer #2).

The reviewers have discussed the reviews with one another and the Reviewing Editor has drafted this decision to help you prepare a revised submission.

Summary:

The study by Hesse and Tsao on "Representation of conscious percept without report in the macaque face patch network" presents confirmatory results reported first in seminal studies by Sheinberg and Logothetis from the 90's showing that neurons in infero-temporal (IT) cortex represent the conscious percept in a binocular rivalry paradigm rather than the physical stimulus. This earlier work has inspired neuroscientists for generations, and the present study is a refreshing update in presenting a novel paradigm to study conscious perception without the necessity of active report.

The authors recorded a large set of neurons with a neuropixel prototype and 32 channel probes in parts of the macaque face patch system, specifically patches ML and AM. They devised a novel task during which perceptual switches between a face and an objects were denoted with specific fixation point locations, so that the percept could be tracked without the need to track manual responses. Thereby, any contributions of the motor system to rivalry could be ruled out. This 'no report' paradigm was first established in humans, and then applied to monkeys trained to fixate for periods of time. The rivalry conditions were compared to physical alternations. The authors show compellingly that a large proportion of ML neurons, and the majority of AM neurons follow the percept during rivalry. The perceptual state could also be decoded from population activity.

The main novelty of this study is the innovation of a new paradigm that can be used to study conscious perception in individuals who are simply trained on fixation. The results (both behavioral and recordings) are compelling. While the study confirms that IT neurons can reflect the percept of an individual, it does not show where that percept is generated, that is whether it reflects feedback signals from PFC, or other neural structures (see Kapoor study), or locally generated, stochastic alternations.

1) The population result is nice given the novel neuropixels recordings, but perhaps not surprising given that the perceptual state could be determined from the majority of single units? Were you able to decode the percept also from the 32 channel probes? Perhaps discuss in greater detail what the population analysis adds.

2) Rivalry switches often occur slowly, often on the order of seconds. It would be helpful to include more details regarding the behavior (e.g. length of fixation, duration of stable percepts, frequency of switches). Was the paradigm presented in a trial structure? How long were these trials? Was this different for humans and monkeys? Your time scale for the human studies denotes several seconds, the monkey timescales are typically a few hundred milliseconds, which is very short. As you know, it often takes time to even become aware of a switch. More detail on these issues will be helpful given that the no report task is the major advance of the study.

3) The authors mention that Frassle et al., 2014, used a no-report paradigm, but I'm curious why the authors didn't discuss other rivalry studies that have used no-report paradigm such as Brascamp et al., 2015, Zou et al., 2016, and Xu et al., 2016. These papers used quite different approaches than the current study, but they seem to establish that the neural modulations that accompany rivalry can occur in the absence of a report.

4) What does it really mean when the cells carry information about both perceived and unperceived stimulus (as shown in Figures 4B and 5B)? If face patch neurons behave in a way of multiplexing information as the authors suggested, then what role might they take in the neural circuits underlying the conscious percept? It has been reported that the responses of high-level visual neurons to suppressive stimuli were almost eliminated (Sheinberg and Logothetis, 1997). Could the authors explain their differences and elaborate why information multiplexing in IT neurons happen only when perceptual reports are not demanded?

5) While I think the novel no-report design very smart, I wonder how accurate it is. The inferred percepts might be distorted by many factors, such as piecemeal rivalry. Such paradigm imperfection might weaken the average response modulation in rivalry condition, as you might oversimplify the percept, which in fact is not complete face or object. If so, does the less pronounced modulation during rivalry than in physical condition truly reflect neural representation, or a side effect from mislabeling?

Revisions expected in follow-up work:

How was the ISCAN system integrated with the goggles? What was the quality of eye movement measurements? Did you examine microsaccades that also can contribute to switches?

---

## [Author Response]

[…] The main novelty of this study is the innovation of a new paradigm that can be used to study conscious perception in individuals who are simply trained on fixation. The results (both behavioral and recordings) are compelling. While the study confirms that IT neurons can reflect the percept of an individual, it does not show where that percept is generated, that is whether it reflects feedback signals from PFC, or other neural structures (see Kapoor study), or locally generated, stochastic alternations.

We would like to thank the reviewers for both the compliments and constructive criticism on the manuscript. We have revised the paper incorporating all the feedback and believe that the manuscript is significantly stronger due to this process.

In addition to the valuable revisions suggested by the reviewers, we are happy to announce that we have also performed a new experiment. Even though the reviewers did not require additional experiments, we believe that this addition adds scientific value to the article by directly addressing the outstanding question whether face patches indeed encode the unperceived stimulus, for which we until now only had suggestive evidence:

Decoding the suppressed stimulus

The finding that modulations in the perceptual condition are weaker and responses are distributed less bimodally on a single-trial basis suggested that cells may be multiplexing information about not only the consciously perceived stimulus but also the suppressed, subconscious stimulus. However, since in all previously recorded sessions binocular rivalry stimuli consisted of only two rivalling images, this could not be shown directly. In a new experiment, we therefore used three images, A, B, and C, and presented two different binocular rivalry stimuli made of image pairs (A,B) and (A,C), respectively (Figure 6A). This allowed us to compare trials where A was consciously perceived but the suppressed stimulus was either B or C, i.e., the animal’s conscious perception was the same in both types of trials, and only the suppressed stimulus varied. We asked whether we can decode the suppressed stimulus, i.e., distinguish between trial types (A,B) and (A,C) based on neural responses, where the image name in bold indicates the consciously perceived image, as inferred by eye movements. We performed this experiment while recording from face patch ML with a 64 channel S-probe. The decoding accuracy for distinguishing the two trial types with different suppressed images was 74% (Figure 6B). This indicates that face cells do encode the subconscious stimulus. Do the same cells multiplex information about both the conscious and subconscious stimulus or are there two distinct subpopulations, with one population encoding the conscious stimulus and another encoding the subconscious stimulus? To address this question, we compared modulation indices for the dominant stimulus with modulation indices for the suppressed stimulus for each cell. For the former, we fixed the suppressed stimulus while varying the dominant stimulus, i.e., 𝑀𝐼_𝑑𝑜𝑚𝑖𝑛𝑎𝑛𝑡_ = (𝑅_A**B**_− 𝑅_A𝐂_)/(𝑅_A𝐁_ + 𝑅_A𝐂_ ), and for the latter we fixed the dominant stimulus while varying the suppressed stimulus, i.e., 𝑀𝐼_𝑠𝑢𝑝𝑝𝑟𝑒𝑠𝑠𝑒𝑑_ = (𝑅_𝐀B_ −𝑅_𝐀C_)/(𝑅_𝐀B_ + 𝑅_𝐀C_ ). We found a positive correlation between dominant stimulus modulation indices and suppressed stimulus modulation indices (𝑝 = 1.4 × 10^−6^, Pearson’s 𝑟 = 0.55, 𝑛 = 66 physically selective cells, Figure 6C). This suggests that cells that are strongly modulated by the dominant stimulus tend to be similarly modulated by the suppressed stimulus. Thus, we did not find evidence for separate populations of cells that encode conscious and unconscious stimulus, respectively.

A natural question arising from the decoding accuracy of 74% is whether this could be due to mislabeling by the no-report paradigm. On some trials, the conscious percept may have been mislabeled as (**A**,B) or (**A**,C) and actually have been (A,**B**) or (A,**C**), respectively. In this case, even if cells only encode the conscious percept and not the suppressed stimulus, the decoding accuracy may have been higher than chance because on those mislabeled trials, the decoder successfully discriminated based on a difference in conscious percept. The following calculation addresses this concern: We will estimate the worst-case decoding accuracy increase we could expect from these mislabelings under the null hypothesis that neurons do not encode the suppressed stimulus. Within image pair (A,B), we could decode (**A**,B) vs. (A,**B**), i.e., whether A or B was consciously perceived as in Figure 4D, with 89% accuracy in this session. If we had recorded more neurons, or neurons that were more selective, we would expect a decoding accuracy at least as high. Given the nature of the no-report binocular rivalry paradigm there is physically no difference between trial types (**A**,B) and (A,**B**), and hence any information that the decoder was able to acquire must have come from the difference in conscious percept. Thus, we can use 89% as a lower bound for the estimated accuracy of the no-report paradigm of inferring the correct conscious percept in this session. Under the null hypothesis that neurons only encode the conscious percept, the decoding accuracy for distinguishing (**A**,B) from (**A**,C) for 89% of trials should be chance (since for these trials, the conscious percept is correctly decoded as A). For the remaining 12% of trials, the conscious percept may have been B or C, respectively. Even if the decoder can decode all of these mislabeled trials with 100% accuracy (which is an overestimate), the decoding accuracy across all trials would be at most 89% × 50% + 11% × 100% = 55.5%. So even in the worst-case, the mislabeled trials would not lead to the observed decoding accuracy of 74%. This suggests that face cells do indeed encode the suppressed image.

We have incorporated the new experiment and associated analyses into the revised manuscript. We believe this additional evidence significantly strengthens the paper, and raise it from a mostly confirmatory study to one that challenges the currently dominant concept of how rivalrous stimuli are represented in IT cortex (Figure 6D, Model I versus Model II).

Revisions for this paper:1) The population result is nice given the novel neuropixels recordings, but perhaps not surprising given that the perceptual state could be determined from the majority of single units? Were you able to decode the percept also from the 32 channel probes? Perhaps discuss in greater detail what the population analysis adds.

We agree that the perceptual modulation of single units predicts that perceptual state can be decoded. The decoding analysis is a proof of concept that perceptual content can be decoded on a single-trial basis with accuracies much higher than chance (95% for physical and 78% for perceptual on average across sessions). This is something we could not achieve with single electrodes: When performing the decoding with single neurons, decoding accuracies were merely 61% ± 12% for physical and 55% ± 6% for perceptual (mean accuracy ± standard deviation across neurons). While the use of Neuropixels prototypes represents an innovation in terms of number of simultaneously recorded channels, it was not necessary to use Neuropixels to decode the percept above chance. Indeed, of the 12 data points in Figure 4D, only three sessions included Neuropixels data. On the 9 other sessions, we recorded with two 32-ch. S-probes in ML and AM and still obtained decoding accuracies much higher than chance. We make this clearer in the text now:

“Recordings were performed using tungsten electrodes (FHC) with 1 MΩ impedance and, after correct targeting was confirmed, with 32-channel S-probes (Plexon) with 75 µm and 100 µm inter-electrode distance, and, in three sessions, with passive Neuropixels-like probe prototypes (IMEC) (Dutta et al., 2019; Jun et al., 2017; Trautmann et al., 2019).”

What the population analysis really adds in our opinion is that we can ask how conscious percepts are encoded during binocular rivalry on single trials. Previous

electrophysiological studies averaged across trials (e.g., Sheinberg and Logothetis, 1997) and found weaker modulation on average for binocular rivalry as compared to physical switches. However, it is unclear whether this weaker modulation strength was the case across all trials or whether it arose from mislabeling of percept on some trials. Therefore, a common perception is that in IT most cells reflect conscious perception exactly (see, e.g., reviewer comment #4 below). Our single-trial analysis of large numbers of simultaneously recorded cells shows that the distribution of single-trial responses during binocular rivalry is less bimodal and spans a smaller range, indicating that cells are truly more weakly modulated during binocular rivalry. This raised the interesting possibility that cells may multiplex information about the veridical physical stimulus and the conscious percept, which we were able to confirm by decoding the suppressed stimulus from a population of simultaneously recorded neurons. We emphasize this fact in the Introduction:

“In a second innovation, we performed electrophysiological recordings using a novel 128-electrode site Neuropixels-like probe that allowed us to measure responses from large numbers of cells simultaneously. […] Inter-trial averaging confounds these two possibilities; to distinguish them, it is critical to compare perceptual versus physical response modulations for single trials.”

2) Rivalry switches often occur slowly, often on the order of seconds. It would be helpful to include more details regarding the behavior (e.g. length of fixation, duration of stable percepts, frequency of switches). Was the paradigm presented in a trial structure? How long were these trials? Was this different for humans and monkeys? Your time scale for the human studies denotes several seconds, the monkey timescales are typically a few hundred milliseconds, which is very short. As you know, it often takes time to even become aware of a switch. More detail on these issues will be helpful given that the no report task is the major advance of the study.

Thank you for this helpful comment. We have now clarified and supplemented the pertaining information in the Materials and methods section. The binocular rivalry stimuli were presented continuously, but fixation spot positions changed at regular intervals and we defined a trial structure based on that. For monkey experiments, the duration of each trial duration was 800 ms (i.e., fixation spots jumped to a new position every 800 ms). For the human experiment, we set the trial duration to 2000 ms, since the study participants had not been extensively trained on the task unlike monkeys and hence needed more time to saccade to the jumping fixation spots. We have now clarified these details in the text:

“Subsequently, for the main experiment, stimuli contained one or two fixation spots at one of four possible locations (top, bottom, left, and right, 1 degree from the center) and were presented for 800 ms ON time and 0 ms OFF time. […] During the binocular rivalry condition, even though the same stimulus was presented continuously, we refer to the 800 ms duration, after which the two fixation spots would change position, as one trial.”

“For human subjects, stimuli were identical except that the trial duration was 2000 ms, since they had not been extensively trained on the task unlike monkeys and hence needed more time to saccade to the jumping fixation spots.”

The trial duration determines the temporal resolution with which we were able to infer switches in percept. However, the trial duration was significantly lower than the average switching time of the percept: In monkeys, median dominance duration was 7.2 seconds for faces and 7.2 seconds for objects. In humans, median dominance duration was 8 seconds for faces and 10 seconds for objects as estimated from fixation patterns, and 8.1 seconds for faces and 8.3 seconds for objects as estimated from reports. We now include the dominance durations in the Results section of the manuscript:

“To account for individuals’ eye dominance, we balanced the contrasts of the stimuli in the two eyes so that the monkey followed both fixation spots equally often in the rivalry condition. […] Similarly, in human subjects median dominance durations were 8 seconds for faces and 10 seconds for objects as estimated from fixation patterns, and 8.1 seconds for faces and 8.3 seconds for objects as estimated from reports.”

3) The authors mention that Frassle et al., 2014, used a no-report paradigm, but I'm curious why the authors didn't discuss other rivalry studies that have used no-report paradigm such as Brascamp et al., 2015, Zou et al., 2016, and Xu et al., 2016. These papers used quite different approaches than the current study, but they seem to establish that the neural modulations that accompany rivalry can occur in the absence of a report.

We thank the reviewers for directing our attention to these interesting alternative approaches to no-report paradigms and have added them to the Discussion:

“Alternative approaches to the no-report paradigms of Frässle et al., 2014, have been developed in which the monkey or human subject is unaware of when a perceptual switch is happening and hence cannot report it, either due to anesthesia or due to the difference in stimuli being too subtle to report. […] Thus, to the best of our knowledge, the current study reveals representation of the conscious percept in IT cells in the most confound-free way to date.”

4) What does it really mean when the cells carry information about both perceived and unperceived stimulus (as shown in Figures 4B and 5B)? If face patch neurons behave in a way of multiplexing information as the authors suggested, then what role might they take in the neural circuits underlying the conscious percept? It has been reported that the responses of high-level visual neurons to suppressive stimuli were almost eliminated (Sheinberg and Logothetis, 1997). Could the authors explain their differences and elaborate why information multiplexing in IT neurons happen only when perceptual reports are not demanded?

It would be hard to imagine how a circuit mechanism within IT could generate switches of conscious percept if IT cells did not encode any information about the suppressed stimulus, since the neural state would be indistinguishable from that to an unambiguous stimulus. The additional experiment described above confirms that cells do indeed multiplex information about the perceived and unperceived stimulus, as both the perceived stimulus and the unperceived stimulus can be decoded from the population, using different decoders. Figure 6C suggests that there are not two distinct populations for encoding perceived and unperceived stimulus, respectively, but the same neuron may have mixed selectivity for perceived and unperceived stimulus. This leaves open the possibility that IT or downstream areas are involved in switches of conscious percept. We mention this in the Discussion:

“To directly test this hypothesis, we presented more than one binocular rivalry stimulus, created from pairs of three images, and found that the subconscious stimulus could indeed by decoded from face patch activity. […] It remains an open question where and how the conscious percept is ultimately isolated from the suppressed stimulus to produce conscious awareness of the former and not the latter.”.

It is a common misconception, which we also had at the outset of this project, that high level visual neurons in IT reflect conscious percept exactly as physical stimuli. Figure 5 of the original paper by Sheinberg and Logothetis, 1997, on neural correlates of binocular rivalry, shows that cells were significantly more weakly modulated during rivalry. Notably, when the non-preferred stimulus was perceived in rivalry, responses were not eliminated. The original study used single electrodes and averaged across trials, and hence, it could not be determined whether the weaker modulation stemmed from mislabeling on a subset of trials. Importantly, we do not think that the multiplexing happens only if reports are demanded. Instead, the weaker modulation appears to be a hallmark of rivalry whether it is reported or not.

5) While I think the novel no-report design very smart, I wonder how accurate it is. The inferred percepts might be distorted by many factors, such as piecemeal rivalry. Such paradigm imperfection might weaken the average response modulation in rivalry condition, as you might oversimplify the percept, which in fact is not complete face or object. If so, does the less pronounced modulation during rivalry than in physical condition truly reflect neural representation, or a side effect from mislabeling?

The effect of mislabeled and mixture trials is a valid concern. We therefore optimized the stimulus to enhance competition between the stimuli and decrease periods of mixture using a variety of methods including: (1) having the stimulus as small as possible while allowing accurate tracking of fixation patterns (5 degree total), (2) increasing contrast of both eyes’ object images, (3) adding fixation marks to help with fusion, (4) adding orthogonal gratings in the background of the objects to increase local orientation contrast, and (5) applying orientation filters to the object images that were orthogonal in left and right eyes to further increase local orientation contrast. We asked human subjects to report whether and how frequently they perceived mixture during the experiment and all subjects reported that they could see only one of the objects most of the time. See the Materials and methods section:

“During the binocular rivalry condition, even though the same stimulus was presented continuously, we refer to the 800 ms duration, after which the two fixation spots would change position, as one trial. […] Moreover, we applied orthogonal orientation filters (with concentration 𝜎_𝑎𝑛𝑔𝑙𝑒_ = 0.5°) to the face and object stimuli, respectively, to increase local orientation contrast and further reduce periods of mixture.”

We think that trials with mixture or mislabeled percept did contribute to the weaker modulation averaged across trials assuming that proportions of mislabeling and mixture were similar. However, we think that these factors cannot explain the radically different response single-trial response profiles between rivalrous and unambiguous conditions; the former were much less bimodal than the latter and spanned a smaller range, despite the binocular rivalry condition having been presented in many more trials. We performed simulations of the effect of mixture on the data shown in Figure 5. We assumed different proportions of mixture from 0%-100% and simulated the worst-case effect of mixture (i.e. exactly half-face, half-object) on responses in the physical condition, by averaging the responses to pairs of face and object trials. We used the same statistical test as described in the paper and found that only if we added 50%-70% mixture to the physical trial responses, did they become statistically indistinguishable from binocular rivalry responses, whereas each human subject reported not seeing any mixture on most trials. Note that this analysis is independent of correct labeling of conscious percepts. We now describe this simulation in the Results section:

“Importantly, this difference in response profiles between physical and perceptual conditions was apparent even when pooling across both face and object trials (Figure 5B, middle), and hence cannot be explained by mistakes in inferring the percept from eye movements. […] Yet, under the reasonable assumption that they were similar, trials with mixed or piecemeal percepts cannot account for the difference in response distributions between physical and perceptual conditions.”

Revisions expected in follow-up work:How was the ISCAN system integrated with the goggles? What was the quality of eye movement measurements? Did you examine microsaccades that also can contribute to switches?

The ISCAN camera recorded the position of one eye through the anaglyph filter. The presence of the filter only slightly impaired the quality of eye movement measurements. We have added this information to the Materials and methods section:

“Eye position was monitored using an eye tracking system (ISCAN). The camera recorded one eye through the red/cyan anaglyph filter.”

We measured the precision of ISCAN eye positions by computing the absolute value of distances between 1 ms adjacent eye data. The median and 99% confidence interval, respectively, were 0.038 degrees and 0.34 degrees. Note that saccades should not contaminate the confidence interval estimate of this jitter, since saccades happen less frequently than every 10 ms. For comparison, the distance between fixation spots was 1.4 or 2 degrees, much larger than the jitter magnitude.

**Author response image 1. sa2fig1:** Quality of eye movement measurements. Histogram shows counts of Euclidean distances between eye positions of adjacent milliseconds in the range from 0 to 1 degree visual angle across all recorded sessions. Median and 99% confidence interval (CI) is shown in orange and yellow, respectively.

We now mention this in the Materials and methods:

“We measured the precision of ISCAN eye positions by computing the absolute value of distances between 1 ms adjacent eye data. […] Note that these confidence intervals should not be contaminated by saccades which occur less frequently than 10 Hz and therefore make up less than 1% of the distribution.”

Given the measurement noise above, we were able to detect saccades over distances of 0.5 degrees or larger, which under some definitions can still be considered microsaccades. It has been reported that perceptual switches happen more frequently around microsaccades (Sabrin and Kertesz, 1980; van Dam and van Ee, 2006). However, in our no-report paradigm we infer the conscious percept based on which fixation spot a subject is saccading to at a given trial, and therefore we can infer percepts with at most the sample rate of saccades. Hence, we cannot determine whether the percept switched more during microsaccades than during static fixation. In terms of neural modulation, we did find that saccades evoked response increases during binocular rivalry, and the response increase was slightly higher when we inferred that the preferred stimulus was perceived compared to the non-preferred, see Figure 5 and Results section:

“We observed response modulations for both physical and perceptual conditions starting around 130 ms after saccade onset (Figure 5A). […] As a consequence, during rivalry the response difference to a saccade between face and object, though significant (𝑝 = 6 × 10^−23^, two-sample t-test, 𝑁 = 701 saccades for object, 𝑁 = 703 saccades for face), was weaker than during the physical condition.”

**References:**

Sabrin, H. W., and Kertesz, A. E. (1980). Microsaccadic eye movements and binocular rivalry. Perception and psychophysics, 28(2), 150-154.

van Dam, L. C., and van Ee, R. (2006). Retinal image shifts, but not eye movements per se, cause alternations in awareness during binocular rivalry. Journal of vision, 6(11), 3-3.